# Differential Impact of Ad Libitum or Intermittent High-Fat Diets on Bingeing Ethanol-Mediated Behaviors

**DOI:** 10.3390/nu11092253

**Published:** 2019-09-19

**Authors:** Nuria Del Olmo, M. Carmen Blanco-Gandía, Ana Mateos-García, Danila Del Rio, José Miñarro, Mariano Ruiz-Gayo, Marta Rodríguez-Arias

**Affiliations:** 1Department of Health & Pharmaceutical Sciences, Facultad de Farmacia, Universidad CEU–San Pablo, Campus de Montepríncipe, 28668 Madrid, Spain; nolmo@ceu.es (N.D.O.); ddrio@ceu.es (D.D.R.); ruigayo@ceu.es (M.R.-G.); 2Department of Psychology and Sociology, University of Zaragoza, C/Ciudad Escolar s/n, 44003 Teruel, Spain; 3Department of Psychobiology, Facultad de Psicología, Universitat de Valencia, Avda. Blasco Ibáñez, 21, 46010 Valencia, Spain; ana.mateos@uv.es (A.M.-G.); jose.minarro@uv.es (J.M.)

**Keywords:** binge drinking, high-fat diet, cognition, anxiety, gene expression, leptin

## Abstract

Background: Dietary factors have significant effects on the brain, modulating mood, anxiety, motivation and cognition. To date, no attention has been paid to the consequences that the combination of ethanol (EtOH) and a high-fat diet (HFD) have on learning and mood disorders during adolescence. The aim of the present work was to evaluate the biochemical and behavioral consequences of ethanol binge drinking and an HFD consumption in adolescent mice. Methods: Animals received either a standard diet or an HFD (ad libitum vs. binge pattern) in combination with ethanol binge drinking and were evaluated in anxiety and memory. The metabolic profile and gene expression of leptin receptors and clock genes were also evaluated. Results: Excessive white adipose tissue and an increase in plasma insulin and leptin levels were mainly observed in ad libitum HFD + EtOH mice. An upregulation of the Lepr gene expression in the prefrontal cortex and the hippocampus was also observed in ad libitum HFD groups. EtOH-induced impairment on spatial memory retrieval was absent in mice exposed to an HFD, although the aversive memory deficits persisted. Mice bingeing on an HFD only showed an anxiolytic profile, without other alterations. We also observed a mismatch between *Clock* and *Bmal1* expression in ad libitum HFD animals, which were mostly independent of EtOH bingeing. Conclusions: Our results confirm the bidirectional influence that occurs between the composition and intake pattern of a HFD and ethanol consumption during adolescence, even when the metabolic, behavioral and chronobiological effects of this interaction are dissociated.

## 1. Introduction

Adolescence is a developmental period in which individuals are highly exposed to environmental risks, such as stress, drug abuse or inadequate dietary habits [1,2,3]. Obesity prevalence is rising in developed countries, particularly among adolescents [4,5,6]. This health problem may be partially due to overexposure to highly palatable and energy-dense food, which promotes hedonic overeating [7,8]. Under particular conditions, binge eating, a type of overeating characterized by an uncontrolled and intermittent food-intake pattern of palatable food, is a phenomenon displayed by many teenagers [9]. Dietary factors are important effectors in the brain, suggesting a direct relationship between obesity and anxiety disorders, motivational disorders or cognitive impairment, although the way in which this diet is ingested determines the extent of these behavioral changes. Animal models confirm that consuming a high-fat diet (HFD) for a long period of time triggers notable cognitive deficits, especially in spatial performance [10,11,12]. In fact, we recently reported a number of consequences that HFDs have on learning and memory processes [13,14] and reward deficits [15,16,17,18]. In a recent study, we observed that memory and spatial learning deficits were only observed in animals with continuous access to a HFD, and after 15 days of HFD abstinence these deficits disappeared. However, mice that binged on fat did not show these deficits [19], and evidence that memory impairment is not related to alleged HFD-induced depression or to anxiety-like behaviors has also been found [12].

Epidemiological studies support a bidirectional positive correlation between the ingestion of palatable food and ethanol (EtOH), specifically among adolescent subjects [20,21], who usually combine EtOH drinking and palatable food during the weekends [21,22,23]. EtOH is the most commonly consumed drug in our society, producing a wide range of behavioral effects and being one of the first drugs of choice among teenagers [24]. Binge drinking, a specific way of consuming alcohol, is a prevalent pattern of alcohol intake during adolescence [25,26], consisting of the consumption of four drinks for women or five drinks for men in about 2 h, leading to a blood EtOH concentration of 0.08 g/dL [27]. EtOH intake during adolescence can produce short- and long-term consequences, such as memory impairment and neural cell death in several brain regions, inducing long-lasting effects that may interfere with normal brain functioning during adulthood [27,28,29,30]. In addition, substance abuse in the early stages of life is linked to a higher rate of drug abuse and dependence in adulthood [31,32].

Despite these previous results, the effects of the interaction between an HFD and EtOH on anxiety and cognitive function remain to be elucidated. In a recent series of basic research, an increased sensitivity to cocaine and EtOH has been described in rodents exposed to an intermittent and limited HFD [16,17,19,33]. However, no attention has been paid to the impact that EtOH + HFD consumption during the juvenile period may have on learning processes and mood disorders. 

The present study evaluates, for the first time, the behavioral profile of adult animals that have been exposed during adolescence to EtOH binge drinking combined with an HFD. Taking into account previous studies modeling binge-drinking behavior [34,35], intermittent EtOH administration was administered to adolescent mice that had been exposed to either a standard diet (SD), continuous access to an HFD (ad libitum HFD), or limited and intermittent access to an HFD (Binge HFD). We sought to establish an eventual correlation between behavioral impairment and biochemical parameters associated with energy metabolism. As there is a debate concerning the role that clock genes could have in connecting drug abuse-related behaviors and obesity/metabolic syndrome [36,37], the expression of genes encoding CLOCK, BMAL1, PER2 and CRY1 were also analyzed within the hippocampus (HIP) and the prefrontal cortex (PFC). 

## 2. Materials and Methods 

### 2.1. Animals 

A total of 90 albino male mice of the OF1 outbred strain (60 for the first and 30 for the second set) were acquired commercially from Charles River (Barcelona, Spain). Animals were 21 days old on arrival at the laboratory and were all housed in groups of 5, under standard conditions (cage size 28 × 28 × 14.5 cm), for 5 days prior to initiating the experimental feeding schedule, at a constant temperature (21 ± 2 °C), with a reverse light cycle (white lights on 19:30–7:30).

Food (SD or HFD) and water were available ad libitum in all the experiments (except during the behavioral tests). All procedures involving mice and their care complied with national, regional and local laws and regulations, which are in accordance with Directive 2010/63/EU of the European Parliament and the council of 22 September 2010 on the protection of animals used for scientific purposes. The Animal Use and Care Committee of the University of Valencia approved the present study. 

### 2.2. Drugs 

Ethanol (EtOH) was administered intraperitoneal (i.p.) at 1.25 g/kg and diluted in 0.9% NaCl at a volume of 0.01 mL/g. 

### 2.3. Procedure 

#### 2.3.1. Feeding Conditions

Two different types of diet were administered in the study. SD (Teklad Global Diet 2014, 13% kcal from fat, 67% kcal from carbohydrates and 20% kcal from protein; 2.9 kcal/g) was given to the control group and a HFD (TD.06415, 45% kcal from fat, 36% kcal from carbohydrates and 19% kcal from protein; 4.6 kcal/g) was administered either ad libitum for the HFD ad libitum group or in a limited way to the HFD binge group. This limited access administration is based on the Corwin et al. [38] model, in which non-food-deprived animals with sporadic and limited access to a HFD develop binge-type behaviors. Both diets were supplied by Harlan Laboratories Models, SL (Barcelona, Spain) and will be referred to from now on as SD, ad libitum HFD, or binge HFD for the sporadic limited access to the HFD. Individual constituents of the HFD (TD.06415): casein (245 g/kg), L-cystine (3.5 g/kg), corn starch (85 g/kg), maltodextrin (115 g/kg), sucrose (200 g/kg), lard (195 g/kg), soybean oil (30 g/kg), cellulose (58 g/kg), mineral mix (43 g/kg), calcium phosphate dibasic (3.4 g/kg), vitamin mix (19 g/kg), choline bitartrate (3 g/kg), red food color (0.1 g/kg). Fatty acid profile (% of total fat): 36% saturated, 47% monounsaturated, 17% polyunsaturated.

On postnatal day (PND) 25, mice were randomly divided into groups (*n* = 15/condition) with similar average body weight (BW) (15–20 g) and assigned either to SD, ad libitum HFD or binge HFD (2 h access on Monday, Wednesday and Friday, following 3–4 h after the beginning of the dark phase). All groups except the ad libitum HFD group were fed with the SD in their own cages. Three days a week, binge HFD groups were exposed to a 2 h fat binge session in a different plastic cage. All the behavioral tests were performed before the binge session, to match the daily conditions throughout the experiment and in order to not interfere in the behavioral outcomes. Water was freely available at all times. Animals were weighed every Monday, Wednesday and Friday throughout the study, and their intake of the SD in their home cage was also measured. 

#### 2.3.2. Drug Pre-Exposure Procedure

The acclimatization period lasted for 7 days (PND 28), and then adolescent animals received 16 doses of EtOH (1.25 g/kg) or saline over a 2-week period according to the following schedule: twice daily administrations (with a 4 h interval) on two consecutive days separated by an interval of 2 days during which no injections were administered. Injections took place on PNDs 28, 29, 32, 33, 36, 37, 40 and 41. These drug administrations were intended to simulate the binge pattern observed in human adolescents and young adults [39]. The i.p. route was chosen in order to ensure that all animals received the same amount of EtOH; in addition, this administration accelerates the process of absorption and is less stressful for the animals than forced oral administration. Behavioral tests were performed 5 days after pre-exposure to EtOH had finalized (PND 46). When binge eating and EtOH injections concurred on the same day, EtOH injections were always administered before each binge eating session.

#### 2.3.3. Experimental Design

An overall and more detailed description of the sets of animals and experimental procedure is provided in Figure 1. Six groups were employed in this experiment: SD + Saline, *n* = 15; SD + EtOH, *n* = 15; Binge HFD + saline, *n* = 15; Binge HFD + EtOH, *n* = 15; Ad libitum HFD + saline, *n* = 15; and Ad libitum HFD + EtOH *n* = 15. EtOH pre-exposure and binge-eating sessions were conducted at the same postnatal period. Binge-eating sessions started on PND 27 and ended at the end of the behavioral testing (PND 55). A total of 13 binge sessions were performed. Within this time period, EtOH pre-exposure started on PND 28, and finalized before the behavioral procedures, which started on PND 46 in the following order: elevated plus maze, object recognition test, passive avoidance test.

A different set of OF1 mice (*n* = 30; SD + Saline, *n* = 5; SD + EtOH, *n* = 5; Binge HFD + saline, *n* = 5; Binge HFD + EtOH, *n* = 5; Ad libitum HFD + saline, *n* = 5; Ad libitum HFD + EtOH *n* = 5) was employed to monitor BW and to analyze plasma biochemistry (triglycerides, glucose, insulin and leptin) and gene expression (leptin receptor, *Lepr*; circadian locomotor output cycles kaput, *Clock*; period circadian protein homolog 2, *Per2*; brain and muscle ARNT-like 1, *Bmal1*; cryptochrome 1, *Cry1*). After dietary/EtOH treatments (PND55), liver, perirenal white adipose tissue (Per-WAT) and subcutaneous WAT (Sc-WAT) were weighed, and the HIP and PFC dissected and processed for gene expression quantification.

### 2.4. Apparatus

#### 2.4.1. Elevated Plus Maze

The elevated plus maze (EPM) consisted of two open arms (30 × 5 × 0.25 cm) and two enclosed arms (30 × 5 × 15 cm) with a central platform (5 × 5 cm) formed by the junction of the four arms. The floor of the maze was made of black Plexiglas and the walls of the enclosed arms of clear Plexiglas. The open arms had a small edge (0.25 cm) to provide the animals with additional grip. The entire apparatus was elevated 45 cm above the floor level. Mice were transported to the dimly illuminated laboratory 1 h prior to testing to facilitate adaptation. Each trial began with the subject placed on the central platform facing an open arm, after which it was allowed to explore for 5 min. The maze was thoroughly cleaned with a damp cloth after each trial. The behavior displayed by the mice was recorded automatically by an automated tracking control system (EthoVision 3.1; Noldus Information Technology, Leesburg, VA, USA). The measurements recorded during the test period were time, frequency of entries and percentage of time spent in each section of the apparatus (open arms, closed arms, central platform). An arm was considered to have been visited when the animal placed all four paws on it. Number of open arm entries, time spent in open arms and percentage of open arm entries are generally used to characterize the anxiolytic effects of drugs [40,41].

#### 2.4.2. Object Recognition

The object recognition test was used to assess recognition memory [42]. The object recognition test was performed as described [43]. The apparatus consisted of an open box (24 × 24 × 15 cm) located in a testing room with constant illumination. The objects used were two small river stones (A) and a small non-toxic plastic toy (B), heavy enough to prevent displacement. On the day before the test, the habituation day, mice were allowed to explore the box (with no objects) for 2 min. On the day of test, a training session (T1) was followed by a test session (T2) after a 1-min interval. Each session (T1 and T2) lasted 3 min. 

For T1, mice were placed in the middle of the box facing away from the two identical stones (AA) arranged in the center of the testing box for 3 min. After this, mice were removed from the box and returned to their home cages. One of the stones was changed to one small toy (non-familiar object). After the retention interval of 1 min outside the testing box, mice were reintroduced into the box for T2. Object exploration was defined as the orientation of the animal’s snout towards the object, within a range of 2 cm or less from the object. Running around the object or sitting on it was not recorded as exploration. Objects were washed with ethanol after each individual trial to equate olfactory cues. The basic measures in the object recognition test were the times spent by the animals to explore an object during T1 and T2 [44]. In addition, e1 and e2 were measures of the total exploration time of both objects during T1 and T2, respectively. The basic measure in the object recognition test was the discrimination index, calculated as DI = ((tnovel − tfamiliar)/(tnovel + tfamiliar)) × 100.

#### 2.4.3. Passive Avoidance Test

A step-through inhibitory avoidance apparatus for mice (Ugo Basile, Comerio-Varese, Italy) was employed for the passive avoidance test. This cage is divided into two compartments (15 × 9.5 × 16.5 cm each one) and was made of Perspex sheets. One compartment is white and illuminated by a light fixture (10 W) fastened to the cage lid (safe compartment), whereas the other is dark and made of black Perspex panels (“shock” compartment). An automatically operated sliding door at floor level divided the two compartments. The floor was made of 48 stainless steel bars with a diameter of 0.7 mm and situated 8 mm apart.

Passive avoidance tests were carried out essentially following the procedure described in Aguilar et al. [45]. On the day of training, after a 60-s period of habituation with each mouse placed in the illuminated compartment facing away from the dark compartment, the door leading to the dark compartment was opened. A foot shock (0.5 mA, 3 s) was delivered when the animal had placed all four paws in the dark compartment. Then, the mouse was immediately removed from the apparatus and returned to its home cage. The step-through latency (time taken to enter the dark compartment) was recorded. Retention was tested 24 h and 6 days later following the same procedure but without the shock. The maximum step-through latency was 300 s.

### 2.5. Plasma Biochemistry 

After dietary and EtOH treatments, the animals (5 per group) were killed by decapitation at 14:00. Blood was collected in cold polyethylene tubes coated with disodium ethylenediaminetetraacetic acid (EDTA), then centrifuged and plasma frozen until assay for plasma leptin and insulin analysis by Enzyme Immunoassay (EIA) (Phoenix Pharmaceuticals Inc., Karlsruhe, Germany; and Mercodia, Sweden, for leptin and insulin, respectively). Glucose was measured spectrophotometrically (Biolabo, Maizy, France).

### 2.6. Quantitative Real-Time PCR

Gene expression was analyzed at 14:00. Total RNA was extracted following the TRI-Reagent protocol (Sigma, St. Louis, MO, USA). cDNA was then synthesized from 2 µg total RNA by using a high-capacity cDNA reverse transcription kit (Applied Biosystems, Foster, CA, USA). Quantitative RT-PCR was performed by using assay-on-demand kits (IScript Advanced cDNA synthesis kit, Bio-Rad, Madrid, Spain) for Lepr (Mm00440181_m1), Clock (Mm00455950_m1), Per2 (Mm00478113_m1), and designed primer pairs (Integrated DNA Technologies, Clarkville, IA, USA) for Bmal1 (forward 5′-ACTGACTACCAGTTAGAATATGCAG-3′; reverse 3′-ATTTTGTCCCGACGCCTCTT-5′) and Cry1 (forward 5′-CTGAAGGAGTGCATCCAGG-3′; reverse 3′-GGCATCAAGATCCTCAAGACA-5′). TaqMan Universal PCR Master Mix (Applied Biosystems) and SsoAdvanced Universal SYBR Green Supermix (Biorad) were used, respectively, for amplification following the manufacturer’s protocols in an ABI PRISM 7000 Sequence Detection System (Applied Biosystems). Values were normalized to the housekeeping gene actin (Rn00667869_m1) or 18S (Rn03928990_g1). The ∆∆C (T) method was used to determine relative expression levels. Statistics were performed using ∆∆C (T) values.

### 2.7. Statistics

Data relating to BW and HFD binge intake were analyzed by means of a mixed analysis of variance (ANOVA) with two between-subjects variables: diet, with three levels (Standard Diet, Ad libitum HFD and Binge HFD), and treatment, with two levels (Saline and EtOH); and a within variable: days, with 7 levels (PND 27, 32, 36, 41, 46, 50 and 55). 

An ANOVA was performed for each measure in the EPM and object recognition, with two between variables: diet, with three levels (SD, Ad libitum HFD and Binge HFD), and treatment, with two levels (Saline and EtOH). For the passive avoidance test, an additional ANOVA was performed with the same two between variables and one within variable: days, with three levels (training, 24-h test and 6-days test). A Bonferroni correction was used for post-hoc comparisons. 

For gene expression and plasma biochemistry analysis, a two-way ANOVA with two between factors; diet, with three levels (Standard Diet, Ad libitum HFD and Binge HFD); and Treatment, with two levels (Saline and EtOH), followed by appropriate post hoc tests (Student–Newman–Keuls and Mann–Whitney U tests) was used. All values are expressed as means ± SEM.

## 3. Results

### 3.1. Ad Libitum HFD, but Not Binge HFD, Increased Body Weight

BW (F(6,504) = 1325.278; *p* < 0.001) increased across the treatment (*p* < 0.001) (Figure 2a). However, ad libitum HFD mice treated with EtOH (F(12,504) = 2.601; *p* < 0.002) displayed higher BW than the other groups (*p* < 0.01 on days 46, 50 and 55). Moreover, as seen in Figure 2b, ad libitum HFD groups (F(2,12) = 70.431; *p* < 0.001) ingested significantly more kcal per day than the other groups (*p* < 0.001). Figure 2c illustrates the escalation in HFD intake during binge episodes, which started on PND27. The ANOVA (F(6,168) = 9.429; *p* < 0.001) revealed that the magnitude of the binge increased from the second binge session on PND32 regardless of alcohol consumption (*p* < 0.001).

### 3.2. Binge HFD Increased Adiposity Levels and Plasma Leptin Only in Mice that Consumed Ethanol 

Table 1 summarizes the effect of dietary and EtOH treatments on the weight and plasma parameters of both organs/tissues. Only the combination of EtOH + ad libitum HFD was able to increase the amount of perirenal (F(2,24) = 5.421; *p* < 0.01) and subcutaneous white adipose tissue (WAT) (F(2,24) = 4.270, *p* < 0.02), which is congruent with the increase of plasma leptin (F(2,24) = 4.739, *p* < 0.01) concentration detected in this group (*p* < 0.01 for perirenal and subcutaneous WAT; *p* < 0.001 for leptin). The liver weight was also increased in the EtOH + Ad libitum HFD group (F(2,24) = 4.905, *p* < 0.01), but in this case the statistical difference was only identified when compared to the EtOH + SD cohort (*p* < 0.05). Finally, plasma insulin concentration (F(2,24) = 8.863, *p* < 0.001) was also higher in the EtOH + Ad libitum HFD cohort (*p* < 0.001).

### 3.3. Mice that Binge on HFD Showed an Anxiolytic Profile

The results of the EPM are presented in Table 2. Animals that binged on HFD, regardless of EtOH administration, spent more time (F(2,84) = 12.815; *p* < 0.001), and percentage of time in the open arms (F(2,84) = 9.195; *p* < 0.001), and making more total (F(2,84) = 14,740; *p* < 0.001) and open entries (F(2,84) = 14,575; *p* < 0.001) than the rest of the groups (*p* > 0.001). There was also a significant effect of diet on the total distance traveled (F(2,84) = 20.786; *p* < 0.001), as animals that binged on a HFD traveled more cm than those on the standard diet and ad libitum HFD groups (*p* < 0.001).

### 3.4. HFD Counteracts the Impairment in the Object Recognition Test Induced by EtOH 

During the first training session (E1) (see Table 3), the time spent exploring the objects (F(2,80) = 6.939; *p* < 0.002) was specifically decreased by the HFD when it was consumed in binge episodes, independently of the consumption of EtOH. However, EtOH decreased the discrimination index data (DI) (F(1,80) = 5.210; *p* < 0.025) only in mice that consumed the SD (*p* < 0.001). 

### 3.5. EtOH Affected Memory in the Passive Avoidance Test

As illustrated in Figure 3, all the groups (F(2,168) = 30.169; *p* < 0.001) increased the latency to enter the dark compartment in the 24 h test session and in the test performed 6 days later (*p* < 0.001 in all cases); although 6 days later, the latency to enter the dark compartment was significantly shorter that in the 24 h test (*p* < 0.01). In addition, EtOH treatment produced a significantly shorter time to enter the dark compartment in the tests that occurred at 24 h and 6 days later (*p* < 0.001 in both cases), and this impairment is not affected by EtOH administration.

### 3.6. Ad Libitum HFD Increased Lepr Gene Expression within the Hippocampus and the Prefrontal Cortex

Leptin receptor gene (Lepr) expression was upregulated in the ad libitum HFD groups in both the HIP (F(2,23) = 8.195, *p* < 0.01) and PFC (F(2,23) = 31.282, *p* < 0.001) areas (*p* < 0.01 in all cases) (Figure 4).

### 3.7. Ad Libitum HFD Altered Clock Genes Expression within the Hippocampus and the Prefrontal Cortex

In the hippocampus, the regulation of *Clock* and *Per2* expression (Figure 5a,b) was dependent on the interaction between EtOH and HFD consumption ((F(2,23) = 5.987, *p* < 0.01) for *Clock*; and (F(2,23) = 4.183, *p* < 0.02) for *Per2*). Specifically, in absence of EtOH, ad libitum HFD repressed *Clock* expression (compared to both control and binge HFD cohorts; *p* < 0.001). Nevertheless, in animals that consumed EtOH, both ad libitum and binge HFD repressed the expression of this gene (*p* < 0.001). In contrast, *Bmal1* and *Cry1* expression remained unaltered (Figure 5c,d).

In the PFC, *Clock* expression was not modified by either EtOH or by HFD intake (Figure 6a), but *Bmal1* (Figure 6c) underwent regulation by HFD that was independent of alcohol intake (F(2,23) = 21.229, *p* < 0.001). In this case, an upregulation of gene expression was specifically observed in mice that consumed ad libitum HFD (*p* < 0.001).

In the case of *Per2* expression within the hippocampus, the influence of EtOH was not significant. Nevertheless, ad libitum HFD (F(2,23) = 5.987, *p* < 0.01) induced *Per2* expression (*p* < 0.01), while binging on a HFD repressed the expression of *Per2* in the absence of EtOH (*p* < 0.01) (Figure 5b). Similar results were obtained in the PFC, where *Per2* (F(2,23) = 11.258, *p* < 0.001) was also upregulated by ad libitum HFD both in saline- and EtOH-treated animals (*p* < 0.01) (Figure 6b). In regard to *Cry1*, the expression of this gene displayed a similar pattern of expression to *Per2* with a significant upregulation in animals that consumed HFD ad libitum (*p* < 0.01) (F(2,23) = 7.117, *p* < 0.01).

## 4. Discussion

Over the past few decades, the world has witnessed the emergence of an obesity pandemic [46] that has co-occurred alongside a gradual increase in worldwide rates of alcohol abuse, specifically in the form of binge drinking [25,26]. Our results highlight the relationship between an HFD and EtOH administration as a complex one, as it depends on the way that the HFD is administered and the variable studied. EtOH binge administration potentiated the effects of ad libitum HFD on body weight, adiposity and insulin and leptin levels. However, bingeing on an HFD during adolescence did not induce appreciable behavioral changes, with the exception of an anxiolytic profile, and was not affected by co-occurrence of EtOH bingeing. Conversely, the detrimental effects of an ad libitum HFD were potentiated by co-exposure to EtOH. On the other hand, HFD, either ad libitum or intermittently, was capable of blocking the detrimental effects of EtOH administration in the object recognition tests observed one week after the last EtOH injection. Changes in the expression of clock genes were mainly due to the ad libitum HFD, independently of EtOH administration. Studies combining both high-fat diets and alcohol binge drinking ingestions during adolescence are scarce, despite the fact that they could enable more focused programs to promote healthier food behaviors in the young population. A recent health survey study [47] showed that the Western diet, based on refined grains, processed meats, and animal fats, is associated with a higher alcohol intake.

With respect to feeding behavior and BW, our current findings are in agreement with other studies showing that intermittent access to an HFD leads to fat-bingeing behaviors [16,38,48,49], lacking BW effects [17]. Overweight and WAT enlargement were mainly observed in ad libitum HFD + EtOH mice, as already reported by Gonçalves and co-workers [50], even after discontinuation of EtOH administration. Accordingly, ad libitum HFD + EtOH bingeing also enhanced plasma insulin despite the absence of plasma glucose variation, suggesting that EtOH bingeing would favor the deleterious effect of the HFD in terms of carbohydrate metabolism. This hypothesis requires further study aimed at characterizing glucose management in this cohort.

After HFD interventions, plasma leptin usually correlates positively with adiposity [17,51]. Here, we observed that the group exposed to ad libitum HFD + EtOH displayed higher plasma leptin levels than the other experimental groups, indicating that bingeing on EtOH also potentiated the effect of HFD in terms of leptin production, coherently with the enlargement of fat pads detected in this group. Interestingly, Lepr expression appeared to be upregulated within the HIP and PFC by EtOH in ad libitum HFD mice, a result that would suggest an increased sensitivity to leptin in these brain areas. Therefore, our results strongly suggest that chronic and intermittent EtOH administration enhances the effects of ad libitum HFD on energy balance. It is worth mentioning that bingeing on an HFD did not induce any disturbance in this respect, independently of EtOH consumption.

In our study, bingeing on an HFD increased the time spent in open arms in the EPM, which could suggest a reduction in anxiety. There are controversial reports regarding the anxiety profile of mice consuming an HFD, as some studies show that these diets promote anxiety-like behaviors [10,52], while others suggest an anxiolytic effect [53,54]. Previous studies carried out in our laboratory demonstrated that continuous access to an HFD does not affect anxiety levels [12,17]. The anxiolytic profile (EPM) displayed by mice bingeing on an HFD differs from those reported in previous studies [17,55] dealing with longer periods of exposition to HFD bingeing (approximately seven weeks). Therefore, shorter exposure to HFD binges could exert a temporary anxiolytic effect that disappears as the binge behavior endures. In agreement with our results, Sirohi and co-workers [56] have recently reported that rats receiving intermittent access (24 h, twice weekly for six weeks) to an HFD show an anxiolytic effect. Bake and co-workers [57] employed locomotor activity and rearing as markers of food anticipatory activity, which we hypothesize could be related to the increase in the time and percentage of time spent in open arms and in the distance traveled in the EPM.

A striking finding in this study was the lack of effect of HFD (both ad libitum and binge) on memory performance in the passive avoidance and the object recognition tests, which may be due to the short duration (PND 25–55) of the dietary treatment used in the current study. Previous studies by our group have found that an HFD impaired hippocampus-dependent memory in mice that consumed an HFD (eight weeks) during the transition from adolescence to adulthood [13,14]. Other reports have also demonstrated that a chronic HFD induces deficits in the object recognition test after three months of HFD administration [56]. One could speculate that the lack of effect of HFD on plasma leptin may spare leptin resistance within the hippocampus, a condition that has been shown to impair memory-related mechanisms [14,58,59].

Conversely, we observed that, independently of HFD consumption, mice exposed to EtOH presented deep disturbances on memory retrieval and learning, in line with previous reports showing that chronic and intermittent EtOH administration during adolescence decreases memory performance in the object recognition test [35,60,61]. Surprisingly, HFD (both ad libitum and binge) mitigated the effect of EtOH on the DI in the object recognition test, suggesting that HFD has a protective effect, which could be linked to an interference with EtOH pharmacokinetics, although this circumstance was not observed in a recent study carried out in our laboratory [16]. A protective effect of HFD has also been reported by Holm-Hansen and co-workers [62] in a study. We can rule out a possible interaction between injected ethanol and an HFD, since we have previously shown that ethanol blood levels do not change when the drug is injected to animals exposed to an HFD binge [16] showing that HFD reverses social recognition memory deficiency in Nrg1 mutant female mice.

In regard to the influence of EtOH and HFD on the expression of clock genes, our results showed that ad libitum HFD, but not HFD bingeing, induced robust changes, independently of alcohol consumption. Otherwise, clock genes were differently regulated in the HIP and the PFC. The most striking result was related to the apparent mismatch between *Clock* and *Bmal1* expression detected in HFD animals (both ad libitum and binge). Thus, HFD repressed *Clock* expression within the HIP in mice treated with EtOH as well as in control mice that consumed the HFD ad libitum, while *Bmal1* remained unaltered in this area. Conversely, in the PFC, *Clock* expression was unaltered while *Bmal1* expression was dramatically increased by ad libitum HFD. We would like to highlight that these findings have to be interpreted with caution, as data regarding CLOCK and BMAL1 protein levels are lacking. In any case, this finding is relevant as *Clock* and *Bmal1* form heterodimers capable of activating the promoter region of the repressor *Per1/Per2* and *Cry1/Cry2* genes to induce their transcription. PER2 and CRY1 proteins heterodimerize in the cytoplasm and translocate to the nucleus to interact with *Clock/Bmal1* genes, inhibiting further transcription and activation [63]. Therefore, this feed-back loop may become partially dysfunctional under our conditions leading to the disruption of transcription regulatory mechanisms in which the heterodimer is involved [64]. Interestingly, HIP *Per2* expression appeared to be upregulated in ad libitum HFD groups, while *Cry1* was mostly unaffected. In regard to the effects detected in the PFC, similar comments may be made, although in this case, an upregulation of *Bmal1* and *Cry1* was observed in ad libitum HFD animals, while *Clock* and *Per2* remained unchanged.

These changes may be relevant in the context of our study as clock genes, and particularly *Per2* and *Clock*, seem to modulate specific drug-induced behaviors [37]. Roybal et al. [65] reported that *Clock* disruption induces a behavioral profile including hyperactivity, lower depression-like behavior, lower anxiety, and an increase in the reward value for cocaine. In this line, studies in mammalian models have shown that CLOCK regulates dopamine function and cocaine reward [66]. Our findings show that mice consuming the HFD ad libitum (independently of EtOH) mostly displayed normal behavior, suggesting that, in our experimental model, the behavioral impairment is mostly independent of circadian factors. In any case, although most studies dealing with the effects of HFD on the expression of clock genes have focused on peripheral and hypothalamic effects [67,68,69], our study shows that both the HIP and the PFC seem to be sensitive to this kind of stimuli. Although other authors have demonstrated that clock expression levels appear to be increased in other central nervous system (CNS) areas of mice with HFD-induced obesity [70], as far as we know, our study is the first to show changes in these genes after HFD in the HIP and PFC.

## 5. Conclusions

Our results showed that the effect induced by adolescent exposure to HFD and EtOH depends mainly on the form in which the HFD is administered. We also demonstrated that the metabolic, behavioral and chronobiological effects of this interaction are dissociated. Ad libitum HFD plus EtOH induced profound changes in body weight and adiposity, altering the expression of leptin receptors in the HIP as well. Although most of the behavioral consequences of HFD or exposure to EtOH during adolescence proved to be independent of each other, it is worth mentioning that an HFD can counteract part of the cognitive deficits induced by EtOH. These results confirm and extend previous work showing that bingeing on an HFD during adolescence increases the consumption rate of EtOH via oral self-administration [16]. Therefore, our results confirm the bidirectional influence that exists between the composition of the diet and the pattern of diet and EtOH consumption during adolescence.

## Figures and Tables

**Figure 1 nutrients-11-02253-f001:**
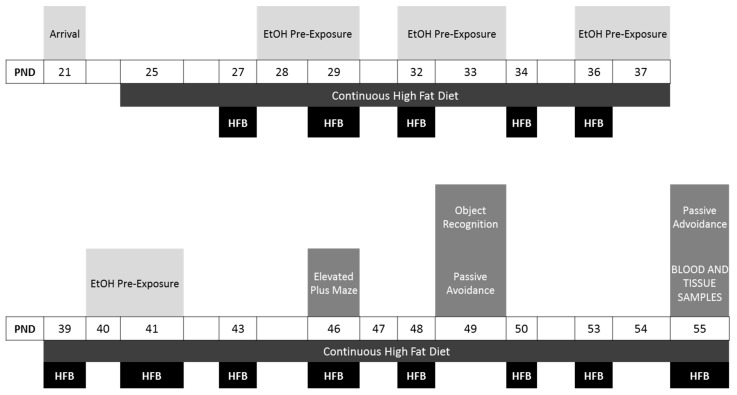
Experimental design. Abbreviations: EtOH, ethanol; HFB,

**Figure 2 nutrients-11-02253-f002:**
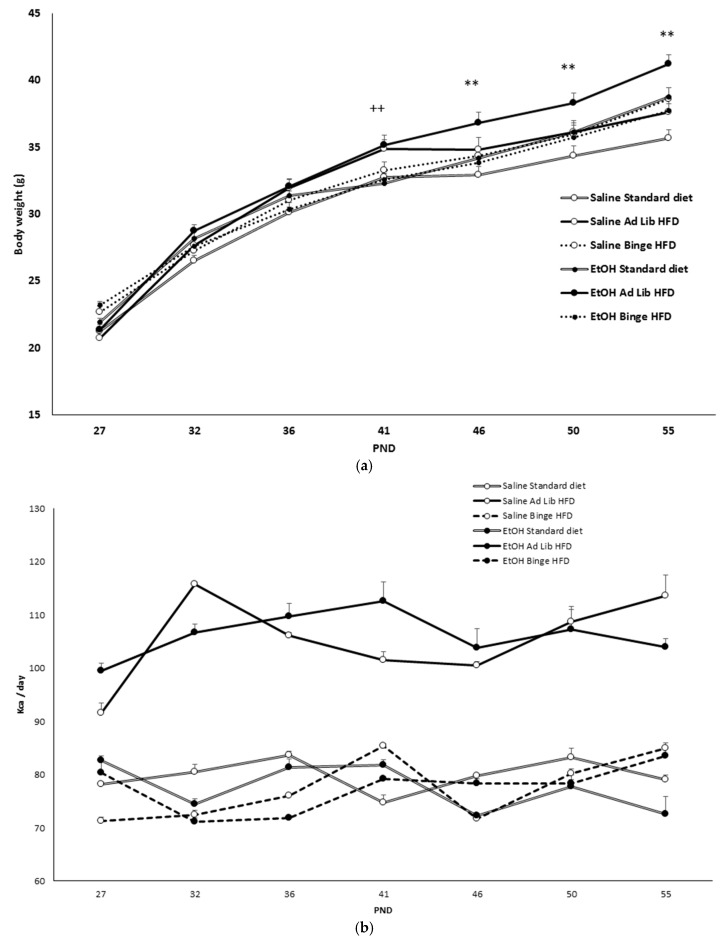
Body weight (BW), daily consumed kcal, and binge intake. (**a**) BW during the whole procedure. (**b**) Daily caloric intake (kcal). (**c**) Mean of binge high-fat diet (HFD) escalation during the whole procedure. ++ *p* < 0.01, significant difference between Ad libitum and Binge HFD groups ** *p* < 0.01, significant difference with respect to the standard diet and the other two ethanol-treated groups; *** *p* < 0.001, significant difference with respect to postnatal day (PND) 27.

**Figure 3 nutrients-11-02253-f003:**
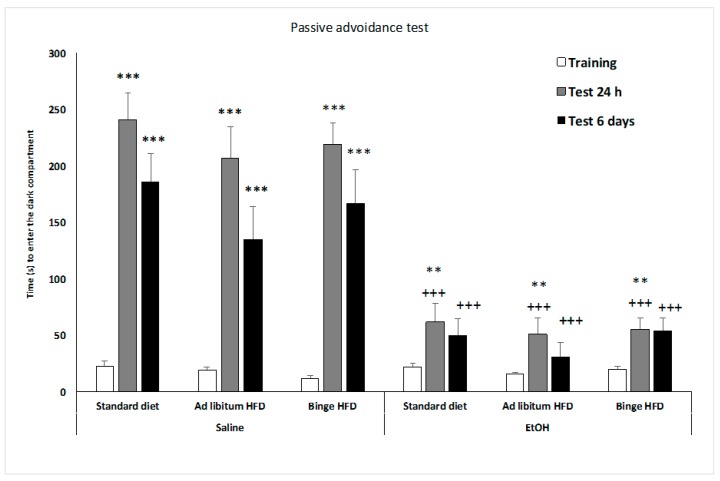
Effect of an ad libitum or binge high-fat diet exposure and EtOH binge on adolescent mice in the passive avoidance test. Bars represent the time taken to enter into the dark compartment in the training and test sessions (24 h and 6 days after training). Data are presented as mean (±SEM), ** *p* < 0.01, ****p* < 0.001 versus training session; +++ *p* < 0.001 versus saline counterparts.

**Figure 4 nutrients-11-02253-f004:**
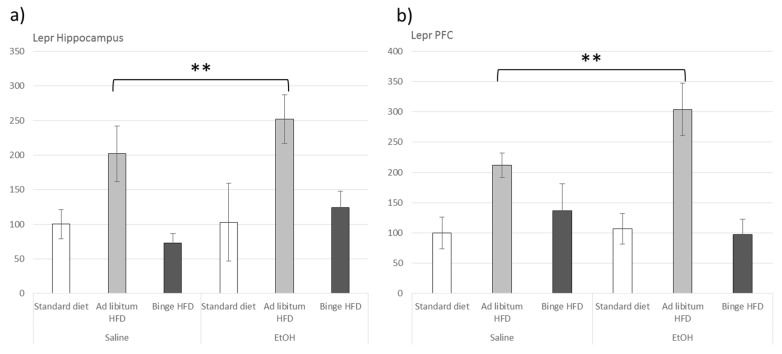
Effect of an ad libitum or binge HFD exposure and EtOH binge on adolescent mice in the leptin receptor expression in the hippocampus (**a**) and the prefrontal cortex (PFC) (**b**). Data are presented as mean (±SEM), ** *p* < 0.01, versus EtOH counterparts.

**Figure 5 nutrients-11-02253-f005:**
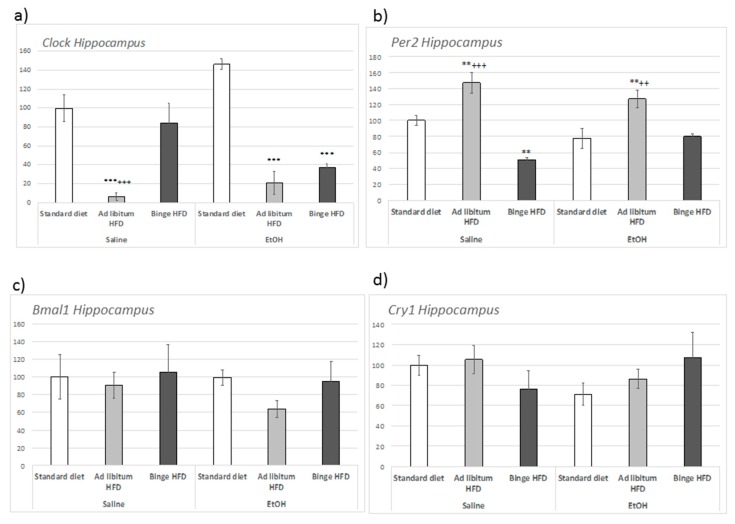
Effect of an ad libitum or binge HFD exposure and EtOH binge on adolescent mice in the circadian gene expression in the hippocampus. (**a**) *Clock*; (**b**) *Per2*; (**c**) *Bmal1;* (**d**) *Cry1*. Data are presented as mean (±SEM) *** *p* < 0.001; ** *p* < 0.01 with respect to the standard diet; +++ *p* < 0.001; ++ *p* < 0.01 with respect to the binge HFD.

**Figure 6 nutrients-11-02253-f006:**
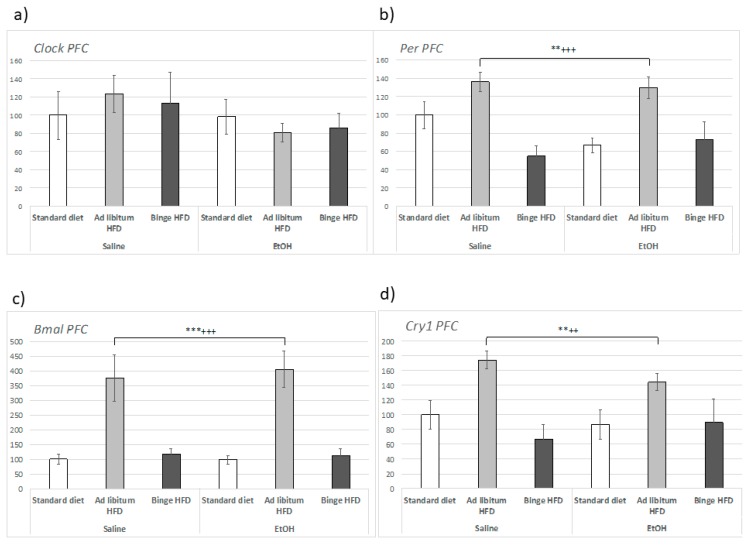
Effect of an ad libitum or binge HFD exposure and EtOH binge on adolescent mice in the circadian gene expression in the prefrontal cortex (PFC). (**a**) *Clock*; (**b**) *Per2*; (**c***) Bmal1*; (**d**) *Cry1*. Data are presented as mean (±SEM) *** *p* < 0.001; ** *p* < 0.01 with respect to the standard diet; +++ *p* < 0.001; ++ *p* < 0.01 with respect to the binge HFD.

**Table 1 nutrients-11-02253-t001:** Effect of dietary and EtOH treatments on tissues weight and plasma. Data are presented as mean (± SEM), + *p* < 0.05 with respect to the corresponding saline groups. ** *p* < 0.01, *** *p* < 0.001 with respect to the rest of the groups.

	Saline			EtOH		
	Standard Diet	Ad Libitum HFD	Binge HFD	Standard Diet	Ad Libitum HFD	Binge HFD
**Liver (g)**	1.83 ± 0.06	1.71 ± 0.12	2.01 ± 0.08	1.75 ± 0.04	1.96 ± 0.08 +	1.79 ± 0.04
**Perirrenal WAT (g)**	0.21 ± 0.01	0.23 ± 0.01	0.21 ± 0.01	0.23 ± 0.01	0.63 ± 0.03 **	0.22 ± 0.02
**Subcutaneous WAT (g)**	0.28 ± 0.05	0.37 ± 0.06	0.35 ± 0.04	0.28 ± 0.06	0.57 ± 0.07 **	0.30 ± 0.03
**Plasma Glucose (mg/dL)**	247 ± 20	193 ± 19	240 ± 19	213 ± 18	220 ± 17	227 ± 15
**Plasma Insulin (ng/mL)**	0.54 ± 0.21	0.29 ± 0.12	0.35 ± 0.07	0.33 ± 0.07	1.17 ± 0.21 ***	0.26 ± 0.05
**Plasma Leptin (ng/mL)**	1.33 ± 0.06	2.18 ± 0.48	1.69 ± 0.24	2.06 ± 0.24	3.98 ± 0.67 ***	1.51 ± 0.23

**Table 2 nutrients-11-02253-t002:** Elevated plus maze. Data are presented as mean (±SEM), *** *p* < 0.001; ** *p* < 0.01 versus standard diet; + *p* < 0.05; ++ *p* < 0.01; +++ *p* < 0.001 versus high-fat diet.

	Saline			EtOH		
	Standard Diet	Ad Libitum HFD	Binge HFD	Standard Diet	Ad Libitum HFD	Binge HFD
**Time in open arms**	58 ± 9	78 ± 12	122 ± 11 **++	76 ± 12	95 ± 10	117 ± 11 **++
**Percentage of time in open arms**	23.9 ± 3.6	32.2 ± 4.9	47 ± 4 **+	33 ± 5	36 ± 3	45 ± 14 **+
**Time in central platform**	52.4 ± 7.2	59.1 ± 8.7	37 ± 5	62 ± 9	58 ± 9	41 ± 5
**Closed Arms**	189.2 ± 11.9	162.6 ± 12.9	139 ± 11	162 ± 15	147 ± 15	142 ± 8
**Entries in open arms**	17.8 ± 1.9	22 ± 2.2	42 ± 5 **++	25 ± 3	23 ± 2	42 ± 5 **++
**Total entries**	50.4 ± 4.1	51.5 ± 4.3	81 ± 6 **++	57 ± 3	51 ± 4	70 ± 8 **++
**Percentage entries in open arms**	35.6 ± 2.9	44.5 ± 4.2	51 ± 4	44 ± 5	45 ± 3	47 ± 3
**Distance traveled**	1688 ± 127	1903 ± 187	3282 ± 386 ***+++	1510 ± 50	1653 ± 77	2890 ± 445 ***+++

**Table 3 nutrients-11-02253-t003:** Object recognition. Data are presented as mean (±SEM), *** *p* < 0.001 with respect to corresponding saline group. DI: discrimination index; E1 and E2: measures of the total exploration time of both objects during test 1 and test 2, respectively.

	Saline			EtOH		
	Standard Diet	*Ad Libitum* HFD	Binge HFD	Standard Diet	*Ad Libitum* HFD	Binge HFD
**DI**	47.8 ± 3.3	36.9 ± 9	34.1 ± 5.1	11.4 ± 6.5 ***	32.4 ± 7.8	33.5 ± 11.2
**E1**	19.8 ± 2.3	27.5 ± 2.4	16.3 ± 3.3	21.5 ± 2.8	26.8 ± 2.8	16.8 ± 3.2
**E2**	26.1 ± 3.1	81.5 ± 5.1	14.7 ± 3.3	21.6 ± 3.1	32.5 ± 5	24.3 ± 9

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
