# Peer review of "Differential Impact of Ad Libitum or Intermittent High-Fat Diets on Bingeing Ethanol-Mediated Behaviors"

_nutrients, 2019, doi:10.3390/nu11092253_

Round 1
Reviewer 1 Report
Summary
Binge alcohol and binge eating frequently co-occur. This paper examined the effect of ad libitum HFD vs. binge HFD relative to ad libitum chow and ethanol binge drinking during adolescence on gene expression of energy metabolism genes and clock genes (PFC, HC), behavioral impairment, and fat composition in young adulthood. Ad libitum HFD plus binge ethanol increased body weight and adipose tissue, plasma insulin, and plasma leptin. Ad libitum HFD increased leptin receptor expression in PFC and hippocampus. Interestingly, HFD protected against ethanol-induced memory deficits. HFD induced an increase in EPM behavior that appeared to be anxiolytic-like. HFD also disrupted Clock and Bmal1 expression. One the major messages of this study was that ad libitum HFD plus ethanol had the most detrimental effects on the measures of health outcome in the mice. Ad libitum HFD access induced obesity and increased insulin and leptin that was potentiated by ethanol. Ad libitum HFD had biggest effects with ethanol. Binge HFD with ethanol had little effect on the various measures. Overall, this is a well-executed, comprehensive study with a well-described experimental design (nice schematic) and appropriate statistical analysis. I have a few minor concerns that could be addressed.
Comments:
What’s the OF1 strain? A few details could be provided. Is this an outbred stock or an inbred strain?
Were females and males used? N=15/condition. Is this mixed females + males? If so, then Sex should be included in the ANOVA model. Was a power analysis performed to decide on this sample size?
More descriptive names/labels could be included for the 6 groups in the figures. While I can make an educated guess, I don’t fully understand the way they are currently denoted. E.g., which groups are HFD binge, HFD ad libitum, and chow controls?
A formal demonstration of desynchronization of clock genes requires several time points. I would recommend caution in using this term based on the current results.
How long after that previous binge episode is EPM assessed? How many hours since the last binge exposure? Recent or are they going through binge “withdrawal”? Without additional measures of anxiety-like behaviors, I would recommend caution in the interpretation that this is an anxiolytic effect. Similar increases in EPM open arm behaviors have been observed during opioid and alcohol withdrawal in mice but I don’t think anyone would argue that drug withdrawal is anxiolytic.
A major limitation of the study is that ethanol is injected rather than self-administered. I’m curious to know if injected ethanol interacts differently with HFD than orally ingested ethanol. Perhaps there is a literature on this? Either way, this caveat could be discussed.
Author Response
Added as word document

Reviewer 2 Report
Summary
This manuscript covers a very important topic about the interaction of ethanol and high fat diet(palatable food) consumption. The study is highly appropriate for this journal. The authors do a nice job of executing a complex 6 group 3 X 2 design with interesting behavioral and gene expression phenotyping. Of note, the alterations in Clock and Per gene expression in the hippocampus are interesting even while only descriptive in nature. The quality of the writing is acceptable with some minor comments below.
Major Comments
- Distance traveled must be reported for EPM. The difference in open arm entries and open arm time might be due to a hyperlocomotion phenotype that is not reported. I think the data are interesting either way, but the authors need to add this in and discuss it. Importantly, the authors will need to distinguish between anxiolysis and locomotor elevation in multiple points of the discussion section if the added data change the interpretation. Another assay like open field or novelty induced suppression of feeding would have been useful here, but the data are the data at this point. The authors should consider using more than just the EPM to phenotype anxiety in the future.
- It isn’t clear from the methods the order of binge HFD and behavioral assays – i.e. are they binging on HFD right before on the same day of running EPM? Interesting discussion point to consider.
- Western blots for many of the clock genes would be a nice addition to the paper to confirm that there are alterations in clock gene protein in addition to message.
- Discussion point to consider: Is INGESTION of alcohol critical to some of the interactions in the clinical literature? Its possible that this particular model in that specific way.
Minor Comments
line 40-41: awkward transition to last clause “able to….”
line 52: Should start sentence with “However,” not “Moreover,”
line 58: Is binge drinking a “new” way of consuming alcohol?
line 254: “with respect to PND 27”
line 259: “coherent?” I think you mean consistent or congruent.
line 289-290: switch “induced” to produced and rephrase to “tests that occurred at 24 hours and 6 days later, and this impairment is not affected by EtOH administration”
line 319: cut “a” from “underwent a regulation”
line 327: change “regulated” to “altered” or something else.
Inconsistent reference to Cry nomenclature in results and figures. Is it Cry(line 310) or Cry1(methods and line 326)? Figures say Cry. Be consistent and correct here
line 341: remove extra comma
line 355-356: Rephrase to “This hypothesis requires further study aimed at characterizing glucose management in this cohort.”
line 419: remove a from “displayed a normal behavior”
Author Response
Added as word document
